# TGFBR3L—An Uncharacterised Pituitary Specific Membrane Protein Detected in the Gonadotroph Cells in Non-Neoplastic and Tumour Tissue

**DOI:** 10.3390/cancers13010114

**Published:** 2020-12-31

**Authors:** Evelina Sjöstedt, Anders J. Kolnes, Nicoleta C. Olarescu, Nicholas Mitsios, Feria Hikmet, Åsa Sivertsson, Cecilia Lindskog, Kristin A. B. Øystese, Anders P. Jørgensen, Jens Bollerslev, Olivera Casar-Borota

**Affiliations:** 1Department of Neuroscience, Karolinska Institutet, Solnavägen 1, 171 77 Solna, Sweden; nicholas.mitsios@ki.se; 2Department of Immunology, Genetics and Pathology, Uppsala University, Dag Hammarskjöldsväg 20, 752 37 Uppsala, Sweden; feria.hikmet_noraddin@igp.uu.se (F.H.); cecilia.lindskog@igp.uu.se (C.L.); olivera.casar-borota@igp.uu.se (O.C.-B.); 3Section of Specialized Endocrinology, Department of Endocrinology, Oslo University Hospital, Rikshospitalet, Pb. 4950 Nydalen, 0424 Oslo, Norway; a.j.kolnes@studmed.uio.no (A.J.K.); n.c.olarescu@medisin.uio.no (N.C.O.); k.a.oystese@medisin.uio.no (K.A.B.Ø.); andjoe@ous-hf.no (A.P.J.); jens.bollerslev@medisin.uio.no (J.B.); 4Institute of Clinical Medicine, Faculty of Medicine, University of Oslo, Box 1072 Blindern, 0316 Oslo, Norway; 5Department of Protein Science, Science for Life Laboratory, KTH-Royal Institute of Technology, Tomtebodavägen 23a, 171 65 Solna, Sweden; asa.sivertsson@scilifelab.se; 6Department of Clinical Pathology, Uppsala University Hospital, 75185 Uppsala, Sweden; 7Department of Pathology, Oslo University Hospital, Oslo University Hospital, Rikshospitalet, Pb. 4950 Nydalen, 0424 Oslo, Norway

**Keywords:** gonadotroph cells, pituitary gland, pituitary neuroendocrine tumours, membrane protein, immunohistochemistry, hormone secretion

## Abstract

**Simple Summary:**

Pituitary neuroendocrine tumours originate from the endocrine cells of the anterior pituitary gland and may develop from any of the cell lineages responsible for producing the different pituitary hormones. The details related to tumour differentiation and hormone production in these tumours are not fully understood. The aim of our study was to investigate an uncharacterised pituitary enriched protein, transforming growth factor beta-receptor 3 like (TGFBR3L). The TGFBR3L protein is highly expressed in the pituitary compared to other organs. We found the protein to be gonadotroph-specific, i.e., detected in the cells that express follicle-stimulating and luteinizing hormones (FSH/LH). The gonadotroph-specific nature of TGFBR3L, a correlation to both FSH and LH as well as an inverse correlation to membranous E-cadherin and oestrogen receptor β suggests a role in gonadotroph cell development and function and, possibly, tumour progression.

**Abstract:**

Here, we report the investigation of transforming growth factor beta-receptor 3 like (TGFBR3L), an uncharacterised pituitary specific membrane protein, in non-neoplastic anterior pituitary gland and pituitary neuroendocrine tumours. A polyclonal antibody produced within the Human Protein Atlas project (HPA074356) was used for TGFBR3L staining and combined with SF1 and FSH for a 3-plex fluorescent protocol, providing more details about the cell lineage specificity of TGFBR3L expression. A cohort of 230 pituitary neuroendocrine tumours were analysed. In a subgroup of previously characterised gonadotroph tumours, correlation with expression of FSH/LH, E-cadherin, oestrogen (ER) and somatostatin receptors (SSTR) was explored. TGFBR3L showed membranous immunolabeling and was found to be gonadotroph cell lineage-specific, verified by co-expression with SF1 and FSH/LH staining in both tumour and non-neoplastic anterior pituitary tissues. TGFBR3L immunoreactivity was observed in gonadotroph tumours only and demonstrated intra-tumour heterogeneity with a perivascular location. TGFBR3L immunostaining correlated positively to both FSH (*R* = 0.290) and LH (*R* = 0.390) immunostaining, and SSTR3 (*R* = 0.315). TGFBR3L correlated inversely to membranous E-cadherin staining (*R* = −0.351) and oestrogen receptor β mRNA (*R* = −0.274). In conclusion, TGFBR3L is a novel pituitary gland specific protein, located in the membrane of gonadotroph cells in non-neoplastic anterior pituitary gland and in a subset of gonadotroph pituitary tumours.

## 1. Introduction

Cell lineage-specific transcription factors of the anterior pituitary gland are important for the development of pituitary cells [1]. The classification of pituitary neuroendocrine tumours (PitNET) [2] is based on their cell lineages and is an important part of diagnosis and treatment planning [3]. The transcription factor SF1 (NR5A1) is specific for gonadotroph cell lineage (FSH/LH), T-Pit (TBX19) for corticotroph cell lineage (ACTH), and Pit-1 (POU1F1) is seen in the somato-, lacto- and thyrotroph cells [1]. PitNETs are of epithelial origin and can develop from any of the pituitary cell lineages. They can produce and/or secrete hormones or be endocrine-inactive [3]. Epithelial-to-mesenchymal transition (EMT) is a process where the tumours lose their epithelial phenotype and develop mesenchymal characteristics [4,5]. A hallmark of EMT is the loss of membranous E-cadherin and the presence of nuclear E-cadherin [6], which is associated with larger and more invasive PitNETs [7,8,9]. Oestrogen receptors α and β (ERα, ERβ) regulate the expression of E-cadherin and seem to influence the clinical course of the gonadotroph tumours [10,11,12]. Somatostatin receptors (SSTR) have been shown to influence PitNETs response to treatment and clinical course in somatotroph and corticotroph tumours [8,9,10,11,12,13]. SSTR3 is the most abundant SSTR in gonadotroph tumours [14]. However, the relationship between tumour differentiation, receptor status and hormone production is not fully understood. 

In the publicly available Human Protein Atlas (HPA) database [15,16], RNA expression and protein localisation data are available for protein-coding genes in human tissues and cells. RNA abundance is used for the classification of genes based on their expression in different types of tissues representing the whole human body. Tissue enriched expression is defined as a 4-fold higher RNA expression in one tissue type compared to the highest expression level in any other tissue. In the current version of the HPA, 26 genes are classified as pituitary gland enriched. Among these, most of the pituitary hormones of the different cell lineages are found, as well as several pituitary specific transcription factors, such as the corticotroph specific T-Pit and the somato-, lacto- and thyrotroph Pit-1. 

Only 3 out of these 26 genes are lacking evidence on the protein level, according to Uniprot protein existence annotation [17], and are thus far only verified at the transcript level. Transforming growth factor beta-receptor 3 like (TGFBR3L) is one of them. The gene *TGFBR3L* is predicted to encode for a 316 amino acid long single-pass membrane protein [18]. The gene name is based on sequence identity (34% positive amino acids) to the C-terminal region of transforming growth factor beta-receptor 3 (TGFBR3), which contains a conserved zona pellucida domain with the potential to bind growth factors [17,19]. TGFBR3, also called betaglycan, has been shown to function as an inhibin co-receptor [20] and detected in gonadotroph cells in the rodent pituitary gland [21,22]. The RNA expression of TGFBR3 in humans was detected across all tissue types and showed no enrichment in the pituitary gland [16]. For TGFBR3L, however, the RNA expression level was found to be 9 times higher in the pituitary gland compared to tissues with the second-highest RNA expression (cerebral cortex and small intestine). 

In a search for additional specific biomarkers for pituitary tumours, we describe here the distribution of TGFBR3L in non-neoplastic anterior pituitary tissue and in a well-characterised cohort of patients with PitNETs. We verified the location of TGFBR3L protein in gonadotroph cells as well as in gonadotroph tumours only. The expression profile of TGFBR3L was heterogeneous within the tumour cell population, and positive cells often displayed a perivascular location. Further, we characterised and correlated the distribution of TGFBR3L to previously known markers of pituitary cell differentiation and EMT in the pituitary tumours. 

## 2. Results

### 2.1. TGFBR3L Tissue Profiling in the Human Protein Atlas

The immunohistochemical (IHC) staining performed within the pipeline of the HPA Tissue Atlas (https://www.proteinatlas.org/ENSG00000260001-TGFBR3L/tissue) suggested that the TGFBR3L protein was present in the human pituitary gland in accordance with the RNA data. Moreover, positive TGFBR3L immunolabeling was restricted to a subset of cells in the pituitary gland (Figure 1). Since protein location was consistent with the RNA expression profile, the anti-TGFBR3L antibody (HPA074356) was marked with enhanced validation [23], thus, according to the HPA standard, it provided evidence on the protein level. 

### 2.2. TGFBR3L is Selectively Detected in a Subset of Gonadotroph Cells in the Non-Neoplastic Anterior Pituitary Gland

In order to verify the localisation of TGFBR3L in the non-neoplastic anterior pituitary cells, cell lineage-specific transcription factors were used. The gonadotroph specific transcription factor SF1 overlapped with the TGFBR3L positive cells, verifying that TGFBR3L positive cells belonged to the gonadotroph lineage. Conversely, we found no overlap with corticotroph transcription factor T-Pit or the somato-, lacto- and thyrotroph transcription factor Pit-1. Figure 2 provides the 3-plex staining of TGFBR3L, together with SF1 and FSH-*β* in the morphologically normal anterior pituitary cells, without pathological change. A similar pattern was observed when combining TGFBR3L, SF1 and LH-*β*. The TGFBR3L fluorescent staining shows a membranous location, similar to what was observed with the chromogenic detection protocol (Figure 1). All cases of the morphologically normal pituitary gland, which was the non-neoplastic area of surgically removed tumour tissue, consistently showed positive membrane staining in a subset of cells (Appendix A). However, we noted that several SF1 positive cells were TGFBR3L negative (Figure 2), indicating heterogeneity regarding TGFBR3L expression within the gonadotroph cell population.

### 2.3. TGFBR3L is Only Detected in Gonadotroph Tumours

We then investigated TGFBR3L in different subtypes of PitNETs in our tissue microarray (TMA) cohort. The same chromogenic IHC protocol, as used for non-neoplastic anterior pituitary (Figure 1), was used for staining of the tumour TMA sections. In total, 230 different tumours were stained (Table 1), and TGFBR3L staining was exclusively seen in the gonadotroph tumours. Of the 110 gonadotroph tumours, 37 (34%) showed positive immunolabelling for TGFBR3L. Figure 3A,B shows the variable staining pattern seen in the gonadotroph PitNETs, ranging from single cells to almost all cells in the TMA core. In total, 29 cases were scored 1 (with less than 10% positive cells), while 3 cases were scored 2 (10–30% positive cells) and 5 cases scored 3 (more than 30% cells stained). The TGFBR3L staining correlated with the TGFBR3L mRNA levels in the 52 tumours available for the mRNA analysis (Table 2). Since many tumours exhibited a low number of positive cells, we also examined several whole tissue sections for intra-tumour heterogeneity. TGFBR3L positive cells tended to show a perivascular orientation (Figure 3C). Among the 10 stained gonadotroph tumours, 2 were negative, 4 with sparse positivity and 4 with >30% cell positivity in whole tumour sections. In addition, 10 non-gonadotroph tumours were stained (4 corticotroph, 3 lactotroph and 3 somato-lactotroph) and showed no positive immunoreactivity in the whole tumour sections. 

### 2.4. TGFBR3L and FSH/LH Staining

Of the 110 gonadotroph tumours included, additional IHC analyses were available for 95 tumours. Of these, 30 gonadotroph tumours were TGFBR3L positive, while 65 were negative. TGFBR3L staining correlated with tumour FSH-β and LH-β staining (Table 2). Interestingly, when examining the intra-tumour heterogeneity for FSH/LH, an occasional perivascular location could also be observed, similar to the TGFBR3L location (Figure 3D). 

### 2.5. TGFBR3L Expression Shows an Inverse Correlation with E-Cadherin and Oestrogen Receptor β

In order to explore a potential role of TGFBR3L in the process of EMT, the relationship between TGFBR3L and E- and N-cadherin was examined. We found an inverse correlation between membranous E-cadherin immunostaining and TGFBR3L staining and mRNA level (Table 2). Additionally, nuclear presence of E-cadherin was more frequently seen in tumours positive for TGFBR3L than in TGFBR3L negative tumours (90% vs. 71%, *p* = 0.039). Neither TGFBR3L immunolabeling nor mRNA correlated with N-cadherin (IHC and mRNA).

We also explored how TGFBR3L correlated with oestrogen receptor (ER) and somatostatin receptors (SSTR), which are additional markers that we have explored previously in gonadotroph tumours [12,13]. ERβ mRNA showed an inverse correlation with TGFBR3L staining, while ERα did not correlate with TGFBR3L on the protein or mRNA level. The measured mRNA levels of SSTR3 correlated strongly to both TGFBR3L mRNA levels and positive immunostaining, whereas SSTR 1, 2 and 5 (IHC and mRNA) did not correlate with TGFBR3L on either level. We did not find any correlation between TGFBR3L positivity and gender, age, tumour size and invasiveness.

## 3. Discussion

We performed an in-depth characterisation of the, thus far, uncharacterised TGFBR3L gene, which shows enriched expression in the pituitary gland, according to the Human Protein Atlas resource. We demonstrated that TGFBR3L is a membrane-bound protein detected in a subset of gonadotroph cells, both in the non-neoplastic anterior pituitary cells and in a subset of gonadotroph PitNETs. Furthermore, we found that TGFBR3L correlated with tumour FSH and LH staining, and 3-plex staining revealed an overlap of TGFBR3L positive cells, SF1, as well as FSH and LH positivity. 

Our result showing that TGFBR3L is a selective marker of the gonadotroph cells is supported by previous findings in mouse [25], rat [26] and developmental human [27] pituitary gland, where single-cell expression analyses highlighted TGFBR3L as highly enriched in the gonadotroph cells. However, this is the first study based on adult human samples as well as the first evidence of existence of TGFBR3L at the protein level. The data in agreement with the analyses performed in rodents indicate a preserved expression across species. Additionally, the selective expression in the human pituitary gland compared to other regions of the brain can be observed in both mice and pigs [16,28]. The detection of heterogeneous protein expression in the non-neoplastic adenohypophysis and within gonadotroph tumours led us to hypothesise the existence of different sub-populations of gonadotroph cells [29]. Further studies are needed to better characterise the heterogeneity, which could also be due to environmental factors in the tumours, especially since we observed a perivascular location of TGFBR3L positive tumour cells.

PitNETs can cause metabolic disorders related to the hypersecretion of pituitary hormones or be clinically non-functioning, with no signs of hormone over-production. Clinically non-functioning PitNETs of gonadotroph lineage are the most common subtype of pituitary tumours, accounting for approximately 80% of all non-functioning PitNETs [30,31]. Functioning gonadotroph tumours are only rarely described clinically and might be underdiagnosed [30]. Gonadotroph tumours are defined by expression of SF1, and in the majority of cases, also present FSH and/or LH, as identified by IHC [32]. Additional transcription factors involved in the differentiation of gonadotroph cells include GATA-2 [1] and GATA-3 [33]. However, these two transcription factors are also involved in the differentiation of TSH producing cells and tumours [1,33]. The function of TGFBR3L is thus far unknown, but the correlation to FSH-β and LH-β, as well as their co-localisation, suggest a gonadotroph–related function, both in the non-neoplastic anterior pituitary gland and in tumour cells. In addition, the correlation observed between TGFBR3L and SSTR3, the gonadotroph associated SSTR [14], further supports the gonadotroph specific function of TGFBR3L. In our cohort of PitNETs, TGFBR3L was positive in slightly more than one-third of the gonadotroph tumours. As our study was based on TMAs, and TGFBR3L positivity shows intra-tumour heterogeneity, we cannot exclude that even a higher proportion of gonadotroph tumours may be TGFBR3L positive. 

Pituitary tumours without IHC detection of pituitary-specific transcription factors or anterior pituitary hormones have been classified as “null-cell adenomas” in the current WHO classification of pituitary tumours [3]. As currently defined, true “null-cell adenomas” represent less than 1% of all pituitary tumours [34]. The existence of this tumour type has been questioned because of its rarity and lack of evidence of the pituitary cell origin [32,35]. A proportion of “null-cell adenomas” may represent other PitNETs, probably gonadotroph tumours that could not be correctly classified due to the methodological aspects, such as pre-analytical problems or suboptimal IHC protocols [32,35]. None of the limited null-cell tumours in the present cohort stained positive for TGFBR3L. Whether TGFBR3L is useful, as an additional marker for characterisation of “null-cell” tumours needs to be clarified in larger cohort of pituitary tumours that fulfils criteria for “null cell adenoma”.

PitNETs are tumours of epithelial origin and may transit to a more mesenchymal phenotype during tumour progression (EMT) [8,9,11]. EMT is marked by loss of membranous E-cadherin [4,5] and, in some cases, nuclear translocation of the protein [6,7]. We observed an inverse relationship between TGFBR3L and membranous E-cadherin staining. Tumours positive for TGFBR3L also had a nuclear accumulation of E-cadherin more frequently. Taken together, this suggests that TGFBR3L is related to down-regulation of E-cadherin and might be involved in mechanisms associated with epithelial-mesenchymal plasticity. 

Oestrogen receptors (ERs) are known to regulate FSH secretion directly on pituitary cells [36] and affect FSH expression in the pituitary gonadotroph cells [37]. In addition, ERs influence the expression of E-cadherin [10,38]. In our study, TGFBR3L correlated inversely with ERβ mRNA levels, but not with ERα at the protein or mRNA level. Since TGFBR3L is related to both ERβ mRNA and FSH-β/LH-β staining, we hypothesise that the protein plays a role in gonadotroph cell differentiation and gonadotropin regulation. Interestingly, TGFBR3, which shows high sequence homology with TGFBR3L [17], functions as a receptor for Inhibin A and supresses FSH production in gonadotroph cells [20]. Additionally, it has been shown that inhibin subunits (mRNA and protein) are present selectively in gonadotroph adenomas and are linked to FSH expression [39,40]. We have recently demonstrated that FSH staining in gonadotroph tumours, similar to TGFBR3L, is associated with lower membranous E-cadherin, increased nuclear E-cadherin and increased ERα staining [41]. Along with the results presented here, this indicates a complex relationship between gonadotroph differentiation and hormone production, which merits further investigation. Lacking functional analysis, the discussion on the possible role of TGFBR3L in gonadotroph differentiation, regulation and tumourigenesis is hypothesis generating. Thus, further mechanistic studies are needed to elucidate the role of this protein in the gonadotrophs. 

Although TGFBR3L shows sequence homology to TGFBR3, we did not investigate its potentially related functions in this study. Despite the lack of evidence, it has often been assumed that there is a related function between TGFBR3L and TGFBR3, solely based on the sequence homology nomenclature strategies [19]. To highlight the TGFBR3L relation to gonadotroph cell biology, we suggest adding ‘Gonadotroph enriched membrane protein’ (GEMP) as a synonym to TGFBR3L. 

Single nucleotide polymorphism (SNP) in TGFBR3L has been associated with the risk of neuroblastoma, especially primary neuroblastoma in the adrenal gland [42]. In the same publication, the authors suggested TGFBR3L to be transcriptionally regulated by N-myc proto-oncogene protein (MYCN), based on the site of the SNP and the correlation between MYCN and TGFBR3L levels. MYCN is a transcription factor associated with oncogenesis [43]. Interestingly, SF-1 has also been mentioned in association with MYCN and neuroblastoma progression [44,45]. The potential role of TGFBR3L in tumorigenesis related to SF-1 needs to be explored in other neoplasms. The association with TGF-beta receptor (although only based on sequence homology), possible regulation by MYCN, a relation to neuroblastoma and the selective detection in gonadotroph non-neoplastic and tumour cells, make this protein relevant for further studies and characterisation. 

## 4. Materials and Methods 

### 4.1. Patient Cohort

The study included 230 PitNETs (Table 1 and Figure 4) operated from 1998 to 2009 at a tertiary referral centre at Oslo University Hospital, Oslo, Norway. None of the patients had previously received radiotherapy for pituitary or brain tumour. For patients with more than one pituitary surgery, only one of the tumour samples was included in the result; additionally, samples with the highest IHC score were selected for positive cases. Informed consent was obtained from all participants. Ethical approval was obtained from the Regional Committees for Medical Research Ethics - South East Norway (REC south-east) and the Oslo University Hospital (REK 2020/24582, approval date 27.06.2014 and 07.04.2020).

### 4.2. Immunohistochemical Characterisation of the PitNET Cohort 

Formalin fixed paraffin embedded tumour tissue was available from all patients. The diagnosis of PitNET was confirmed through haematoxylin and eosin staining, and cell lineage was determined using pituitary hormones and cell lineage-specific transcription factors. Two 1 mm cores from each tumour were used to construct tissue microarrays (TMA), which were used for further analysis, as previously described [46]. 

Gonadotroph tumours, defined by positivity for the transcription factor SF1 and/or FSH/LH, accounted for 110 of the tumours (Table 1). For these tumours, the following IHC analyses were available: Membranous E-cadherin (intracellular domain), SSTR1, SSTR2, SSTR3, N-cadherin in 95 patients, ERα in 93 patients and SSTR5 in 91 patients. The IHC staining details have been described previously for SF-1 [46,47], FSH and LH [12,46], E-cadherin [47], ERα [12], SSTR 1-3 and 5 [13]. 

FSH and LH immunostaining demonstrated variable intensity from weak to strong in gonadotroph tumours and were scored on a scale from 0 to 4 based on the percentage of positive cells; 0 = no positive cells; 1 = 0–10% positive cells; 2 = 10–50% positive cells; 3 = 50–80% positive cells; and 4 for >80% positive cells. IHC positivity for membranous E-cadherin, ERα and SSTRs was quantified using an immunoreactivity score (IRS). The IRS was the product of the percentage of positively stained cells (0 = 0%; 1 = 1–10%; 2 = 10–50%; 3 = 50–80%; and 4 = >80%) and the predominant staining intensity (0: No staining; 1: Weak staining; 2: Moderate staining; 3: Strong staining). Nuclear E-cadherin was considered to be either positive or negative in a binary manner. All previous IHC analyses were performed by OC-B.

### 4.3. Whole Tumour Sections and Non-Neoplastic Anterior Pituitary Gland

In addition to the cohort of PitNETs, TGFBR3L IHC was performed on 20 whole tumour tissue sections (10 gonadotroph tumours and 10 non-gonadotroph tumours; 4 corticotroph, 3 lactotroph and 3 somato-lactotroph). Eight out of these whole tumour sections contained morphologically normal (non-neoplastic) anterior pituitary tissue removed during the pituitary tumour surgery (TGFBR3L IHC images shown in Appendix A). The distribution of positive cells and the staining pattern observed for TGFBR3L IHC were similar in all the samples; only one sample was used for representative multiplex immunofluorescence (3-plex) staining. Whole tumour sections were collected from Uppsala Biobank, anonymised and approved by the Regionala Etikprövningsnämnden (Regional Ethics Review Board) in Uppsala (Reference # 2002-577, 2005-338 and 2007-159, approval dates 20.11.2002, 20.12.2005 and 31.07.2008 respectively). 

### 4.4. TGFBR3L Antibody 

The generation of the anti-TGFBR3L antibody (HPA074356) followed the standardised procedure used within the Human Protein Atlas [15]. The human genome sequence (TGFBR3L was encoded by the human gene ENSG00000260001) was used as a template to design a protein fragment antigen with low homology to other human proteins. A peptide corresponding to 51 amino acids (FPGGLKGSARFLSFGPPFPAPPAPPFPAAPGPWLRRPLFSLKLSDTEDVFP) was designed as an antigen. His tag and ABP were included, and the recombinant protein was then produced in E.coli, purified and used for immunisation. After dual-column solid-phase system purification of the antiserum, the antibody was tested on a protein array for affinity control containing the actual antigen together with 384 other random peptides for specificity testing; one single peak was shown (more information can be found online [48].

### 4.5. TGFBR3L Chromogen Immunohistochemistry

Initial IHC staining was performed within the Tissue Atlas pipeline, with horseradish peroxidase (HRP) polymer conjugated secondary antibody and chromogenic 3,3’-diaminobenzidine (DAB) visualisation in line with a previously described protocol [15], which can be found online [16]. An identical protocol was used for the chromogen IHC on the TMA from the pituitary tumour cohort and the whole tissue sections. Coverslip mounting was performed using Pertex (Histolab, Västra Frölunda, Sweden) following dehydration in alcohol and Tissue clear (VWR). Counterstaining was done with HTXplus (Histolab, Västra Frölunda, Sweden); moreover, all remaining reagents, including antigen retrieval, wash and blocking buffers, HRP polymer, DAB chromogen and substrate were from LabVision (Fremont, CA, USA). Staining protocols were performed in an Autostainer 480 (LabVision) and Leica CV5030 (Leica Biosystems, Nussloch, Germany) and antigen retrieval in a pressure boiler (Decloaking chamber, Biocare Medical, Walnut Creek, CA, USA) using a pH6 citrate buffer (LabVision). The primary antibody (anti-TGFBR3L, HPA074356) was diluted 1:100 from the stock concentration (0.03 mg/mL) and incubated for 30 min at room temperature. Image digitalisation was performed using Scanscope AT2 (Aperio, Vista, CA, USA) using a 20× objective. 

In the TMAs, TGFBR3L staining was scored from 0 to 3: Negative = 0, 1 = less than 10% positive cells, 10–30% cells positive = 2 and score 3 for >30% cells stained positive. TGFBR3L immunoreactivity was assessed by ES and OC-B, who were blinded to the clinical data. 

### 4.6. TGFBR3L Multiplex Fluorescence Immunohistochemistry

To enable multiplex staining using antibodies raised in the same species (rabbit), we chose to use a 3-plex TSA strategy [49], where antibodies were added one at a time, and insoluble tyramide signal amplification (TSA) was used for visualisation followed by heating and inactivation of the first antibody. This was then followed by the second antibody and a different TSA fluorophore (TSA-Plus; PerkinElmer). The same protocol and reagents used for chromogen staining, with 30 min primary antibody incubation and 30 min secondary HRP polymer, were applied, except for 2 additional washing steps (one 5 min wash after primary antibody and one 10 min wash after the secondary antibody). Aqueous mounting media with DAPI (ab104139 Abcam) was used after the final round of staining. Anti-SF1 (ab217317 Abcam) was diluted 1:300 and anti-FSH-*β* (MCA1028 bio-rad) was diluted 1:25,000. The anti-TGFBR3L (HPA074356 atlas antibodies) was diluted 1:300, incubated overnight at 4 °C; and HRP conjugated secondary antibody (P0217 Dako) was used. The protocol for TGFBR3L differed from the other antibodies, based on technical optimisation to achieve a clear membranous staining, similar to the chromogen images, without cytoplasmic spillover. Digital fluorescent images were obtained using a “VSlide” slide scanning microscope (MetaSystems) equipped with a CoolCube 2 camera (12-bit grey scale), a 10× objective. The images (vsi-files) were additionally extracted to high quality .jpeg files for further analysis using the software Metaviewer^®^ (Metasystems). Additional fluorescent immunohistochemical staining was performed using anti-T-PIT (HPA072686, Atlas antibodies, diluted 1:300), anti-PIT-1 (HPA041646, Atlas antibodies, diluted 1:1500) as well as anti- LH-*β* (AB944, Merck, diluted 1:2000).

### 4.7. Real Time-qPCR

Frozen tissue was collected at the operating theatre and stored at −80 °C. mRNA was extracted, and real-time quantitative reverse transcription polymerase chain reaction (RT-qPCR) was performed as described previously [50,51]. Only samples from gonadotroph tumours included in the TMA were examined using RT-qPCR. mRNA analyses of E-cadherin, SSTR2, SSTR3, ERα, ERβ were available from 70 patients, partly described in a previous study [52]. SSTR1 and SSTR5 mRNA data were available for 69 patients, FSH data for 58 patients and TGFBR3L data for 52 patients. 

TGFBR3L mRNA was detected using Forward primer (FP) 5′-GCTGGTGTTGGCAGCCTTC-3′ and Reverse primer (RP) 5′-GCTGGGTGTATCTCCGGACC-3′. ERβ was detected using FP 5′-TTCAAAGAGGGATGCTCACTTC-3′ and RP 5′-CCTTCACACGACCAGACTCC-3; and FSH-β with FP 5′-TGCATAAGCATCAACACCACT-3′ and RP 5′-GGCCTCGCACAGTACAATCAG-3′. The remaining primers used for PCR have been described previously for E-cadherin and N-cadherin [52]; ERα, SSTR1-3 and SSTR 5 [12]. Gene expression was quantified using the delta-delta Ct (∆∆Ct) method and normalised to GAPDH and ALAS1 Ct levels, as they have previously shown to be some of the most stable reference genes in NFPAs, and expressed as relative mRNA levels [51]. 

### 4.8. Statistics

Between-group comparisons were performed using the Mann–Whitney U-test and Chi^2^-test. Spearman’s rank correlation was used for correlation analyses. A *p*-value of <0.05 was considered significant. Stata 16.0 for Windows (StataCorp LLC, College Station, TX, USA) was used for all statistical analyses. 

## 5. Conclusions

TGFBR3L is a recently described pituitary gland specific protein, previously not investigated in the human pituitary. Here, we were able to show the membranous protein location and verify the selective expression in gonadotroph cells in the non-neoplastic anterior pituitary gland and in a subset of gonadotroph pituitary tumours. The function of TGFBR3L is currently unknown, but the results suggest a role in gonadotroph cell development and function based on the correlation with FSH/LH and possibly tumour progression related to the epithelial-to-mesenchymal transition process. 

## Figures and Tables

**Figure 1 cancers-13-00114-f001:**
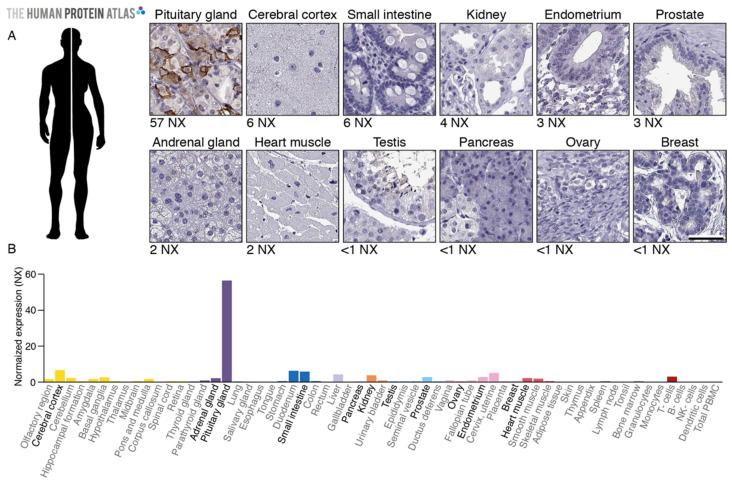
TGFBR3L is only detected in the pituitary gland on the protein level. (**A**) The protein localisation profiled by immunohistochemistry (HPA074356) indicates membranous positivity in subsets of cells in the anterior pituitary gland, while the remaining tissues are negative. (**B**) The RNA level of the gene TGFBR3L in the pituitary gland is reported to be 57 NX (short for Normalised Expression, a normalised version of the transcripts per million [24]), which is 9 times higher than tissues with the second-highest RNA expression (cerebral cortex and small intestine, both 6 NX). The tissues shown in A are indicated in bold font. Images and expression data are from www.proteinatlas.org [16]. Scale bar 50 μm.

**Figure 2 cancers-13-00114-f002:**
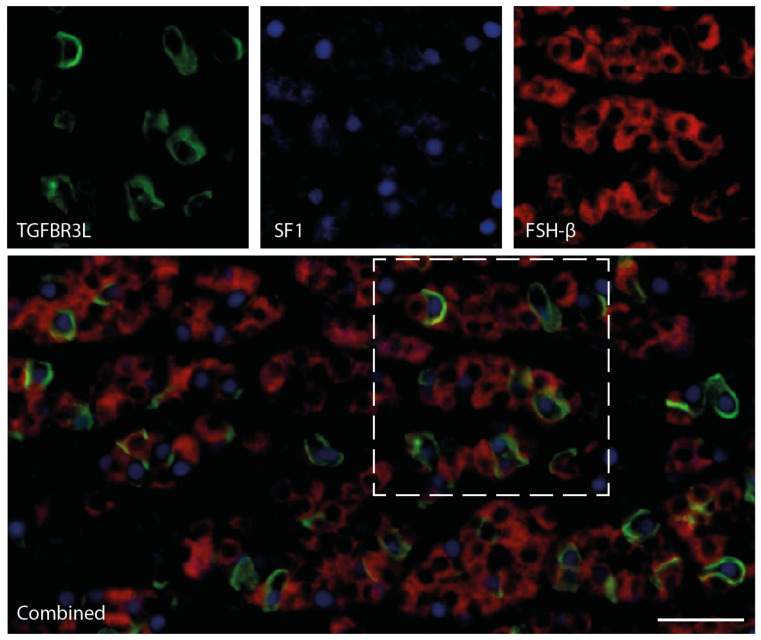
Immunofluorescence staining of TGFBR3L with SF1 and FSH-β in the non-neoplastic anterior human pituitary gland. A 3-plex immunostaining protocol using TSA amplification and heat elution of antibodies was used for the analysis of the three gonadotroph specific proteins: TGFBR3L (green), SF-1 (blue) and FSH-β (red). All TGFBR3L positive cells were also SF-1 positive. The individual markers within the dashed area are shown separately above the combined image. Scale bar 50 μm.

**Figure 3 cancers-13-00114-f003:**
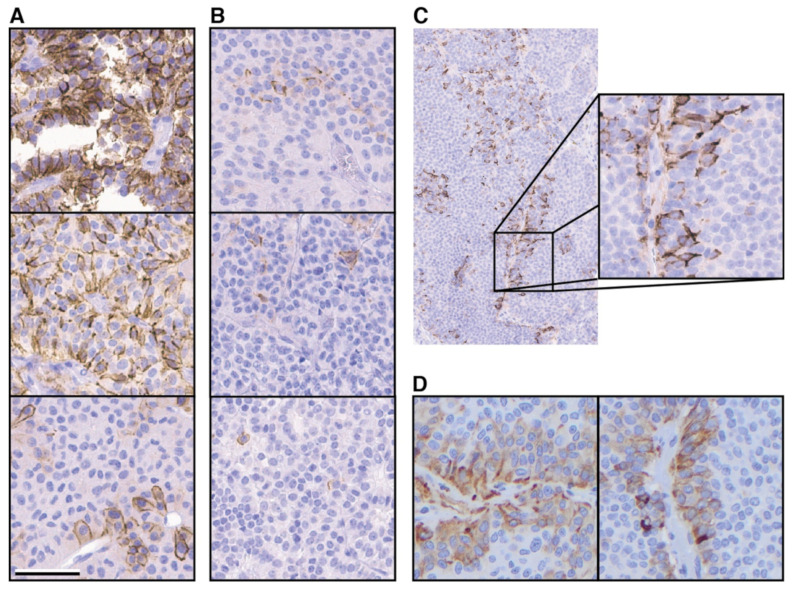
TGFBR3L detection in pituitary tumours. Standard IHC protocol, using chromogen and the TGFBR3L antibody (HPA074356), was applied to a TMA cohort of 230 pituitary tumours and several whole tumour sections. (**A**) Representative images of TGFBR3L staining in gonadotroph tumours scored 2 or 3, with more than 10% (score 2) or 30% (score 3) cells stained positive. (**B**) Examples of staining in gonadotroph tumours scored 1, with less than 10% cells stained positive, represented by only a few cells. (**C**) Example of a whole gonadotroph tumour section stained for TGFBR3L, showing the perivascular location. (**D**) Two examples of similar perivascular FSH-β positivity. Scale bar 50 μm.

**Figure 4 cancers-13-00114-f004:**
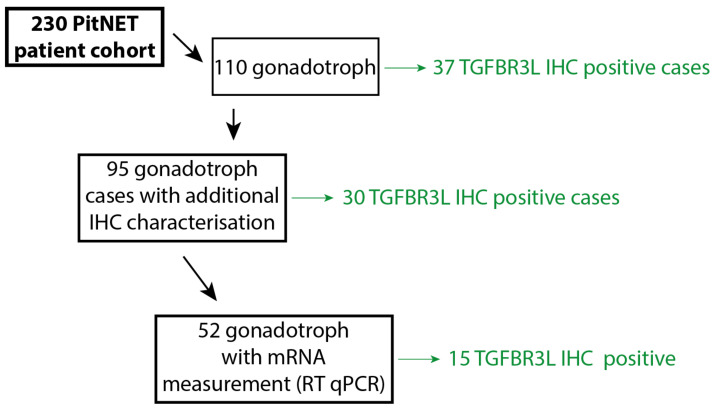
An overview of the number of patients included in the different steps. In total, 230 different cases were annotated in the PitNET cohort, out of which 110 were gonadotroph tumours (Table 1). Of these gonadotroph tumours, 95 cases included additional IHC and mRNA characterisation data from previous studies, used here for correlation analyses, Table 2. Fifty-two cases out of the 95 gonadotroph tumours were available for further TGFBR3L mRNA analysis (RT-qPCR), Table 2.

**Table 1 cancers-13-00114-t001:** Characterisation of the PitNET TMA cohort, including TGFBR3L.

Tumour Type	Number	IHC Subtype	Clinical Phenotype	TGFBR3L Positive
Gonadotroph	110	SF1; FSH/LH	NF-PitNET ^1^	37
Corticotroph	21	T-Pit; ACTH	Cushing/Nelson	0
	25	T-Pit; ACTH	NF-PitNET ^1^	0
Somatotroph	24	Pit-1; GH	Acromegaly	0
	31	Pit-1; GH-PRL	Acromegaly	0
	2	Pit-1; GH	NF-PitNET ^1^	0
Lactotroph	6	Pit-1; PRL	Hyperprolactinemia	0
	2	Pit-1; PRL	NF-PitNET ^1^	0
Thyrotroph	2	Pit-1; TSH	HyperTSH	0
Null cell	7	TFs ^2^ neg; Hormone neg.	NF-PitNET ^1^	0
Total	230			37

^1^ NF-PitNet: Non-functioning pituitary neuroendocrine tumour. ^2^ TF: Transcription factor.

**Table 2 cancers-13-00114-t002:** Correlation analysis of TGFBR3L staining and mRNA compared to gonadotroph hormones, E-cadherin, oestrogen receptors and SSTR3.

Staining/mRNA	TGFBR3L Staining	TGFBR3L mRNA
TGFBR3L mRNA	*R* = 0.378	-
	*P* = 0.0058	-
	*N* = 52	-
FSH staining	*R* = 0.290	*R* = 0.261
	*P* = 0.004	*P* = 0.061
	*N* = 95	*N* = 52
LH staining	*R* = 0.390	*R* = 0.598
	*P* < 0.0001	*P* < 0.0001
	*N* = 95	*N* = 52
E-cadherin IRS ^1^ (membranous)	*R* = −0.351	*R* = −0.387
	*P* = 0.0005	*P* = 0.004
	*N* = 95	*N* = 52
E-cadherin mRNA	*R* = −0.086	*R* = −0.265
	*P* = 0.482	*P* = 0.0598
	*N* = 70	*N* = 51
ERβ mRNA	*R* = −0.274	*R* = −0.221
	*P* = 0.022	*P* = 0.119
	*N* = 70	*N* = 51
SSTR3 IRS ^1^	*R* = 0.049	*R* = −0.218
	*P* = 0.638	*P* = 0.061
	*N* = 95	*N* = 52
SSTR3 mRNA	*R* = 0.315	*R* = 0.818
	*P* = 0.008	*P* < 0.0001
	*N* = 70	*N* = 51

^1^ IRS: Immunoreactivity score.

## Data Availability

The RNA expression profile in human tissues (Figure 1) is based on publicly accessible and downloadable data, available in the HPA Tissue Atlas [16]. The IHC images in human tissues are also available online [16]. The data presented in this study related to the tumour samples is available on request from the corresponding author.

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
