# Peer review of "TGFBR3L—An Uncharacterised Pituitary Specific Membrane Protein Detected in the Gonadotroph Cells in Non-Neoplastic and Tumour Tissue"

_cancers, 2020, doi:10.3390/cancers13010114_

Round 1

Reviewer 1 Report

The authors described carefully the TGFBR3L expression in a large series of PitNETs.

The paper is very clear, well presented and illustrated;

I have raised only one point

The clinical data (age, sex, tumoral size, invasion in cavernous sinus or sphenoidal sinus, hormonal status) should be added to search a correlation with TGFBR3L expression

Author Response

Thank you for the positive feedback and for highlighting a question related to clinical data. We have looked into this and no association was observed between TGFBR3L expression and clinical information (age, sex, tumor size). We have added a sentence to the manuscript to clarify (rows 207-208). 

Unfortunately, the information on tumor size and invasiveness available for the present cohort was limited. We hope to investigate the potential correlation between TGFBR3L and tumour growth further in a future study. 

Reviewer 2 Report

This is an interesting and well-written original article on a new gonadotroph-specific protein, which could see an application in the classification of pituitary neuroendocrine tumors.

The idea to start the article with a simple summary is a nice addition and reads well.

The introduction is precise and contains all required background information for this area.

Placing the Methods sections between discussion and conclusion appears unfavourable. However, if this meets the publication guidelines of Cancers, it is of course fine.

With only this small remark, the manuscript appears ready for publication. It is scientifically sound and the statistics are correct. The successive experiments follow a clear structure and controls are used properly when necessary.

Author Response

Thank you for the positive feedback.

We have followed the publication guidelines for Cancers. 

Reviewer 3 Report

This study describes the distribution within the pituitary of the poorly characterised (but pituitary enriched) protein TGFBR3L. The authors show that the protein is specifically expressed in a sub-set of gonadotrophs and in a proportion of gonadotroph tumours. A varying proportion of cells within gonadotroph tumours are positive for TGFBR3L by immunostaining, which interestingly correlates with loss of E-cadherin membrane staining and other markers of gonadotroph tumours. The study is well designed and the data convincing.

Major comments:

  • the authors suggest that the presence and absence of  TGFBR3L supports a hypothesis of sub-populations of gonadotrophs. This warrants further discussion-in the single cell sequence data that the  authors allude to is there an indication that the TGFBR3L expression is restricted to a sub-population of gonadotrophs that can be defined as a distinct population based on a number of genes? The analysis in tumour tissues would argue against a  sub-population as the dogma is that tumours are clonal, in which case variable TGFBR3L is more likely a result of the local environment rather than differentiation of a distinct sub-lineage of gonadotrophs.The authors may also want to interrogate the data sets of Ho et al (PMID: 32193873) and Zhang et al (PMID: 33077725) that have recently been published.
  • The possible roles of TGFBR3L in gonadotroph differentiation, regulation and tumourigenesis in the discussion is, in my view, overstated. The authors have only described correlation between, for example loss of E-Cad membrane expression and  TGFBR3L. In the absence of any functional analysis, these remain only correlative data and suggesting a role in these  gonadotroph biology are over-speculative.

Reviewer 4 Report

This is a well-conducted and well-witten study, that characterize the expression of TGFBR3L, demontrating, for the first time in humans, that it is restricted to normal gonadotroph cells and gonodotroph Pit-NETs.

I would only suggest to the Authors to briefly expand in the discussion the possible relationships between TGFBR3L and Inhibin A, which has been demonstrated to be selectively expressed in gonadotroph Pit-NETs (Haddad, J Clin Endocrinol Metab. 1994 Nov;79(5):1399-403; Bilezikjian, Mol Cell Endocrinol. 2012 Aug 15;359(1-2):43-52; Uccella, Pituitary. 2000 Nov;3(3):131-9).

In addition, the polishing of English language by a native speacker would be adivisable.

Author Response

Thank you for your suggestion and references on the inhibin/activin system in the gonadotrophs. There is no data on the relation TGFBR3L and Inhibin A. As we have already discussed in the manuscript (please see lines 273-295), there seems to be a relation of TGFBR3 and Inhibin A: “Interestingly, TGFBR3, which shows high sequence homology with TGFBR3L, functions as a receptor for Inhibin A and supresses FSH production in gonadotroph cells [20]”. Based on your input, we have added a sentence with suggested references, in lines 295-296, “Additionally, it has been shown that inhibin subunits (mRNA and protein) are present selectively in gonadotroph adenomas and are linked to FSH expression [39,40].”

We plan in vitro studies in primary gonadotrophs and a mouse gonadotroph cell line to further elucidate the role of TGFBR3L on hormone production in gonadotrops. Until these results are available, we choose to restrain the discussion on the possible relationships between TGFBR3L and Inhibin A.

Round 2

Reviewer 1 Report

I accept your answer on the fact that you can't response to the correlation between TGFBRL3 and clinical phenotype. 

We have to do that in a future study